# Assessing the Pre-Vaccination Anti-SARS-CoV-2 IgG Seroprevalence among Residents and Staff in Nursing Home in Niigata, Japan, November 2020

**DOI:** 10.3390/v14112581

**Published:** 2022-11-21

**Authors:** Keita Wagatsuma, Sayaka Yoshioka, Satoru Yamazaki, Ryosuke Sato, Wint Wint Phyu, Irina Chon, Yoshiki Takahashi, Hisami Watanabe, Reiko Saito

**Affiliations:** 1Division of International Health (Public Health), Graduate School of Medical and Dental Sciences, Niigata University, Niigata 951-8510, Japan; 2Japan Society for the Promotion of Science, Tokyo 102-0083, Japan; 3Niigata City Public Health and Sanitation Center, Niigata 950-0914, Japan

**Keywords:** SARS-CoV-2, nursing home, epidemiology, seroprevalence, pre-vaccination

## Abstract

An outbreak of coronavirus disease 2019 (COVID-19) occurred in a nursing home in Niigata, Japan, November 2020, with an attack rate of 32.0% (63/197). The present study was aimed at assessing the pre-vaccination seroprevalence almost half a year after the COVID-19 outbreak in residents and staff in the facility, along with an assessment of the performance of the enzyme-linked immunosorbent assay (ELISA) and the chemiluminescent immunoassay (CLIA), regarding test seropositivity and seronegativity in detecting immunoglobulin G (IgG) anti-severe acute respiratory syndrome 2 (SARS-CoV-2) antibodies (anti-nucleocapsid (N) and spike (S) proteins). A total of 101 people (30 reverse transcription PCR (RT-PCR)-positive and 71 RT-PCR-negative at the time of the outbreak in November 2020) were tested for anti-IgG antibody titers in April 2021, and the seroprevalence was approximately 40.0–60.0% for residents and 10.0–20.0% for staff, which was almost consistent with the RT-PCR test results that were implemented during the outbreak. The seropositivity for anti-S antibodies showed 90.0% and was almost identical to the RT-PCR positives even after approximately six months of infections, suggesting that the anti-S antibody titer test is reliable for a close assessment of the infection history. Meanwhile, seropositivity for anti-N antibodies was relatively low, at 66.7%. There was one staff member and one resident that were RT-PCR-negative but seropositive for both anti-S and anti-N antibody, indicating overlooked infections despite periodical RT-PCR testing at the time of the outbreak. Our study indicated the impact of transmission of SARS-CoV-2 in a vulnerable elderly nursing home in the pre-vaccination period and the value of a serological study to supplement RT-PCR results retrospectively.

## 1. Introduction

Since the first report of the novel severe acute respiratory syndrome coronavirus 2 (SARS-CoV-2) in Wuhan, China, at the end of 2019, the coronavirus disease 2019 (COVID-19) pandemic has become a threat to public health on a global scale [1]. In Japan, the COVID-19 pandemic started on 16 January 2020, with the first confirmed case being a returnee from Wuhan, China, and the number of infections increasing exponentially from January to April 2020 and gradually spreading to prefectures in Japan, including the Niigata Prefecture. In this context, multiple transmission clusters associated with the “three Cs”—closed spaces with poor ventilation, crowded places with many people nearby, and close-contact settings where many people gather in close quarters—have been identified in nurseries, nursing homes, hospitals, care facilities, schools, and other locations to date [2,3,4,5].

Since most infected individuals remain mild or asymptomatic, it is widely accepted that the volume of unreported cases of COVID-19 is substantial. Several previous studies have described how the proportion of asymptomatic infections reached more than 20.0% in the elderly population in 2020 [6,7]. Indeed, despite many mild cases, some may progress to severe diseases, resulting in an estimated infection fatality ratio of as high as approximately 6.4% in people over 70 years old in 2020 [8,9,10,11]. In this context, nursing home residents are a highly vulnerable population to the spread of SARS-CoV-2 and have accounted for a significant proportion of the virus-induced disease burden during the ongoing pandemic worldwide [12]. Furthermore, because of Japan’s super-aged society, community super spreading occurred from the resident community of older adults, and the transmission was sustained among people in that age group [5]. Although detection of SARS-CoV-2 viral ribonucleic acid (RNA) by real-time reverse transcription PCR (RT-PCR) is generally considered the gold standard for diagnosis of COVID-19, the high proportion of asymptomatic individuals in COVID-19, as noted above, may also underestimate the incidence and prevalence of the disease [13]. Therefore, the surveillance of anti-SARS-CoV-2 immunoglobulin G (IgG) serum antibody titer is one of the useful methods for precise determination of the number of affected individuals in the target population of a community, in addition to detection of the viral genome by RT-PCR [13]. Given this high proportion of unreported infections and the poor prognosis for elderly patients, it is important to conduct serological surveys to gain a complete picture of the COVID-19 disease dynamics and burden in target groups.

In this present study, we aimed to assess seroprevalence almost half a year after the COVID-19 outbreak that occurred in November 2020 in a nursing home in Niigata, Japan, by measuring IgG anti-SARS-CoV-2 antibodies to anti-nucleocapsid (N) and anti-spike (S) proteins among pre-vaccination residents and staff utilizing the enzyme-linked immunosorbent assay (ELISA) and the chemiluminescent immunoassay (CLIA). Seroprevalence numbers will not only provide a measure of the cumulative incidence of SARS-CoV-2 infections but also provide additional insight into the usefulness of comparing anti-N and anti-S antibodies following infection with the virus.

## 2. Materials and Methods

### 2.1. Study Design and Participants

An outbreak of COVID-19 occurred in an elderly nursing home with 97 residents and 81 staff in Niigata, Japan, in November 2020. We conducted a cross-sectional sero-epidemiological study in April 2021 to measure anti-SARS-CoV-2 IgG antibodies (i.e., anti-N and anti-S proteins) for the remaining elderly residents and staff approximately six months after the outbreak. After written informed consent was obtained, blood (serum) was collected from the forearm vein using a winged needle (21G) and EDTA-2Na/F-treated vacuum blood collection tubes (5 mL), and serum samples were centrifuged before measurement and stored in a freezer at −20 °C until testing. Epidemiological data such as age (years), sex (i.e., male or female), and occupation of staff (i.e., doctor, nurse, caregiver, or clerk) were collected. Additional individual characteristics, such as comorbidities and anthropometric measurements, were not collected. All individuals were sampled before the COVID-19 messenger RNA (mRNA) vaccination. 

### 2.2. Measurement of Quantitative Antibody Levels in Serum

The anti-N and anti-S SARS-CoV-2 antibodies were measured by the ELISA method from DENKA (Tokyo, Japan) and the commercial CLIA method from Abbott (Chicago, IL, USA). Specifically, the two immunoassays included are as follows: (i) The ABBOTT SARS-CoV-2 IgG assay (Abbott, Chicago, IL, USA), which is a CMIA for the qualitative detection of IgG antibodies that target the anti-N and anti-S antibodies [14,15,16]. The positive cut-off index was ≥1.4 (S/N ratio) for anti-N antibodies and ≥50.0 AU/mL for anti-S antibodies for the Abbott. These tests were performed using the high-throughput ARCHITECT i2000SR. (ii) The DENKA SARS-CoV-2 IgG assay (DENKA, Tokyo, Japan), which is an ELISA method, is similarly used for the detection of IgG antibodies against N and S antigens using a prototype indirect enzyme immunoassay (DK20-COV4E) [17]. Each well of a 96-well microplate was coated with recombinant SARS-CoV-2 S and N proteins. Serum specimens diluted at a 1:200 ratio with dilution buffer were added to each well. After one hours of incubation at room temperature, the wells were washed three times with washing buffer. Horseradish peroxidase-conjugated goat anti-human IgG antibodies were added to each well, and the plate was incubated at room temperature for 1 h. After five washes, the substrate was added to each well, and the plate was incubated at room temperature. Reactions were stopped by the addition of reaction stopper. Finally, optical density (OD) 450 and OD 630 were measured with a Sunrise™ plate reader (Tecan, Männedorf, Switzerland). Antibody titers were calculated in units of binding antibody unit (BAU)/mL with calibrators assigned to the first World Health Organization (WHO) international standard for anti-SARS-CoV-2 immunoglobulin (National Institute for Biological Standards and Control (NIBSC) code 20/136) [18,19]. The positive cut-off index was ≥30.0 index BAU/mL for anti-N antibodies and ≥50.0 BAU/mL for anti-S antibodies for the DENKA. All tests were performed and interpreted according to the manufacturer’s instructions for each immunoassay, respectively, in a biosafety level 2 (BSL-2) capacity laboratory.

### 2.3. Statistical Analysis

Data were described as the median [interquartile range (IQR)] for continuous variables and frequency (%) for categorical variables. Test seropositivity and seronegativity of the DENKA and Abbott methods for anti-N and anti-S antibodies sampled in April 2021 were calculated based on the results of the RT-PCR that was implemented at the time of the outbreak in November 2020 [20]. A Cohen’s kappa statistic (*κ*) was estimated for each of the anti-N and anti-S antibodies to assess the level of interrater concordance between the two assays of the DENKA and Abbott methods beyond chance. The *κ* coefficient value was classified as slight (0.00 to 0.20), fair (0.21 to 0.40), moderate (0.41 to 0.60), substantial (0.61 to 0.80), and almost perfect (0.81 to 1.00) according to Landis and Koch criteria [21]. Spearman’s rank-order correlation coefficient (*ρ*) was used to investigate the linear associations between anti-N IgG antibodies and anti-S IgG antibodies for each method of DENKA and Abbott. Statistical significance was set at *p* < 0.05, using a two-tailed test. All analyses were performed using EZR version 1.27 [22].

### 2.4. Ethical Consideration

This study was approved by the Niigata University Ethical Committee (approval number 2020–0429) and followed the Declaration of Helsinki (as revised in 2013). Participation in the study was voluntary, and a written informed consent was obtained from each participant.

## 3. Results

In November 2020, a COVID-19 outbreak occurred in an elderly nursing home in Niigata, Japan, with a total of 178 persons, 97 residents, and 81 staff. None of the elderly or staff received COVID-19 vaccines at the time of the outbreak because the COVID-19 mRNA vaccination program in Japan started in February 2021. Active epidemiological investigations initiated on 16 November 2020, when this nursing home informed the Niigata City Public Health and Sanitation Center, Niigata, Japan, that several residents were symptomatic with rapid antigen diagnostic tests positive for SARS-CoV-2. RT-PCR testing for the residents and staff was carried out by the local Public Health and Sanitation Center at the time of the outbreak to identify cases. As a result, a total of 63 cases (attack rate of 32.0%, 63/197), i.e., 56 residents (attack rate of 57.7%, 56/97), and 7 staff (attack rate of 8.6%, 7/81), were identified by RT-PCR. The first round of RT-PCR was performed for all residents and staff on the same and the following days, i.e., Novemebr 16 and 17, when this facility reported the outbreak to the authority. Then RT-PCR was repeated for all residents and staff almost weekly up to December 16, 2020, until no more positive RT-PCR results were identified. All of the residents and staff who were positive with RT-PCR were hospitalized due to the Japanese government’s policy for quarantine purposes during 2020. There were no direct deaths from COVID-19 among residents or staff.

To understand the temporal dynamics of the outbreak, epidemic curves were constructed for 58 of the 63 (92.1%) initial cases (Figure 1A), for whom the date of illness onset was known. The outbreak started on 10 November 2020, when one resident (the index case) developed a fever and two additional residents developed symptoms on the following day. One staff member became symptomatic three days after the onset of the index elderly resident. The epidemic peaked on 14 and 15 November 2020, and ended in two weeks, suggesting the infection may have spread from a single source of exposure in a short time period. This facility is a two-story building, and the initial cases were on the second floor. The attack rate of the elderly on the second floor was higher (85.4%; 41/48) than on the first floor (46.8%; 15/32), suggesting COVID-19 spread quickly in the closed settings and nearly all residents on the second floor were infected. Notably, all five asymptomatic RT-PCR positives were elderly residents and, but no were staff included. As a source of infection, there is a possibility that an asymptomatic staff member that was not detected by RT-PCR may have introduced the virus to this facility. Given that the initial elderly cases were staying in the facility long before the onset of there outbreak, there is no chance of them introducing infections, except for through daily contact with the staff. Although the local Public Health and Sanitation Center obtained demographic information for RT-PCR positive cases, such as age, staying floor for residents, or occupation for staff, no such information was available for RT-PCR negatives.

In the present study, we conducted a cross-sectional sero-epidemiological study in April 2021 for 103 persons (i.e., 41 residents and 62 staff) in a nursing home in Niigata, Japan, approximately six months after the outbreak occurred in November 2020 (Table 1). It should be noted that only 57.8% (103/178) of the initial population and 47.6% (30/63) of RT-PCR positives participated in this study owing to reasons such as declining to participate, leaving or retirement of staff, or deaths of elderly residents that were not related to COVID-19 during the five months after the outbreak. Of these study populations, 9.7% (6/62) of the staff and 58.5% (24/41) of the residents were RT-PCR positive. 

The median age was 49.0 years (IQR: 35.3–55.8) for staff and 90.3 years (IQR: 86.0–94.0) for residents, and most participants were females (≥66.7%) (Table 1). Majority of staff members were caregivers (60.0%, 37/62), followed by nurses (25.0%, 14/62), clerks (11.0%, 11/62), and doctors (4.0%, 4/62). Among staff, most cases were negative for RT-PCR results (90.3%, 56/63), whereas among residents, the positive and negative groups were each approximately half of the cases (58.5% for positive and 41.5% for negative), which was compatible with the initial investigation conducted by the local Public Health and Sanitation Center that COVID-19 infection rates were higher in elderly residents than in staff members. Note that among the staff, only the caregivers were RT-PCR positive, and no other occupations (doctor, nurse, and clerk) were positive. The epidemic curve of the 29 RT-PCR positives who participated in this study (Figure 1B) is almost like the original outbreak of RT-PCR positives (Figure 1A).

For a total of 101 patients, excluding the two who were unable to have serum samples collected, IgG antibody titers were measured (Figure 2). For the staff, the positivity rate of anti-N antibody titers was 9.7% (6/62) for DENKA and 9.7% (6/62) for Abbott (Figure 2A), while those of anti-S antibody were 9.7% (6/62) for DENKA and 12.9% (8/62) for Abbott (Figure 2B). For the residents, the positivity of anti-N antibodies was 48.7% (19/39) for DENKA and 41.0% (16/39) for Abbott (Figure 2C), while those of anti-S antibodies were 56.4% (22/39) for DENKA and 56.4% (22/39) for Abbott (Figure 2D). Overall, these results showed a higher prevalence of positive antibody titers in residents than in staff for both anti-S and anti-N antibodies, consistent with a higher number of infections in residents based on RT-PCR testing during the outbreak.

To quantify the diagnostic performance of the IgG antibody titer assay conducted in this study, the seropositivity and seronegativity of the DENKA and Abbott methods were calculated based on RT-PCR results (Table 2, Appendix A). For DENKA, the seropositivity and seronegativity of anti-N antibodies were 66.7% (20/30) and 93.0% (66/71), respectively, while the seropositivity and seronegativity of anti-S antibodies were 90.0% (27/30) and 97.2% (69/71), respectively. For Abbott, the seropositivity and seronegativity for anti-N antibodies were 66.7% (20/30) and 97.2% (69/71) respectively, while the seropositivity and seronegativity for anti-S antibody titers were 90.0% (27/30) and 95.8% (68/71), respectively. These results show that the seropositivity and seronegativity were almost identical between the DENKA and Abbott methods for anti-N and anti-S antibodies, respectively; however, the seropositivity of anti-N antibodies was relatively lower than that of anti-S antibodies at approximately six months after the outbreak. When divided by residents or staff, the seropositivity and seronegativity for anti-N and anti-S antibodies showed similar tendencies between DENKA and Abbott and were in agreement with the overall results (Appendix A). Notably, there were two RT-PCR-negative but positive anti-S and anti-N antibody titers in one staff member and one resident without any symptoms, suggesting that a small number of asymptomatic infections were missed. Specifically, the staff was a caregiver in this nursing home. This staff’s anti-S and anti-N antibodies were 331.6 AU/mL and 3.55 Index (S/N ratio) for Abbott, and 56.4 BAU/mL and 30.0 BAU/mL for DENKA, respectively, indicating that this staff was seropositive. Meanwhile, the resident showed anti-S and anti-N antibodies that were positive (568.4 AU/mL) and 2.23 index (S/N ratio) for Abbott, and positive 150.9 BAU/mL and 79.6 BAU/mL for DENKA, respectively. Comparing the match between the DENKA and Abbott methods, Cohen’s kappa statistic showed a level of 0.75 (95% confidence interval [CI], 0.59 to 0.90) and 0.87 (95% CI, 0.76 to 0.98) for anti-N and anti-S antibodies, respectively, demonstrating substantial high concordance. In addition, when also divided by residents or staff, the statistics for anti-N and anti-S antibodies showed similar estimates between the two assays (Appendix A). When assessing the association between IgG titers for anti-S and anti-N antibodies, a significant linear correlation was observed for both methods: 0.64 (*p* < 0.01) for DENKA and 0.66 (*p* < 0.01) for Abbott. This indicates that patients with higher anti-S IgG showed higher rates of anti-N antibody positivity, but in turn, patients with lower anti-S IgG tended to show negative anti-N antibodies (Appendix A).

## 4. Discussion

In the present study, we measured anti-SARS-CoV-2 IgG antibodies targeting anti-N and anti-S proteins by two laboratory-based immunoassay testing methods (i.e., DENKA and Abbott) using serum specimens collected from unvaccinated residents and staff of a nursing home in Niigata, Japan. Two main findings were obtained: first, virus transmission spreads within an enclosed environment, and the sero-prevalence in April 2021 among the residents and staff was approximately 40.0–60.0% and 10.0–20.0%, respectively, which was in close agreement with the initial RT-PCR test positive results (57.7% RT-PCR positive for residents and 8.6% for staff) at the time of the outbreak in November 2020, indicating that the infection rates were almost seven times higher among residents. Secondly, seropositivity for anti-S antibodies showed high concordance with RT-PCR positive results even after approximately six months of infections (90.0% for both DENKA and Abbott). The seronegativity for anti-S antibodies also showed high concordance with RT-PCR negatives (97.2% for DENKA and 95.8% for Abbott). Meanwhile, seropositivity for anti-N antibodies remained low at 66.7% for both DENKA and Abbott. It should also be noted that there were two RT-PCR-negative results that had positive anti-S and anti-N antibody titers (one staff and one resident).

In this nursing home, the initial RT-PCR-based infection of elderly residents (57.7%) was almost seven times higher than that of staff (8.6%), indicating a higher risk of infection among the elderly residents, and serological results supported the similar findings. In this study, there were no direct deaths from COVID-19. One impressive paper from January–May 2020 in Japan by Iritani et al. reported that the number and size of clusters in elderly care homes were independently associated with higher mortality rates in all 47 prefectures in Japan, underlining the importance of infection control in such facilities to avoid pressure on local healthcare [23]. Indeed, a report from Nagasaki City, Nagasaki, Japan, showed that although the number of age-specific occurrences per population was not as high for residents of elderly care facilities, deaths were overwhelmingly higher in these facilities (incidence rate of 48.1 per 100,000 person-year) and approximately twice as high as for community-dwelling older adults [24]. In addition, consistent with our present results, a large cohort study of approximately 1500 residents and more than 3000 staff in 201 long-term care facilities in the United Kingdom (UK) between June 2020 and May 2021, measuring anti-N antibodies, found that seropositivity in the last 11 months was 34.6% for residents and 26.1% for staff, suggesting a higher rate of infection among residents [25]. It is imperative to protect care home residents who are vulnerable to COVID-19 infections from potential sources of SARS-CoV-2 through rapid screening and response measures to minimize the scale of outbreaks once the infection is introduced [26].

Interestingly, only caregivers were infected among the staff (doctor, nurse, caregiver, and clerk) in this nursing home. Indeed, one notable paper on healthcare workers in United States of America (USA) hospitals and nursing homes in July–August 2020 showed that seroprevalence also varied by occupation [27]. More specifically, for example, nurses (4.2%) and receptionists/medical assistants (4.1%) were more likely to be seropositive than physicians (2.2%) for hospitals, while in nursing homes, nursing assistants (19.9%) and social workers/case managers/counselors (21.7%) were more likely to be seropositive than occupational/physical/speech therapists (9.8%). Although their studies were not able to explicitly assess the reasons for these differences (i.e., heterogenicity) in seropositivity between occupations, it was observed that occupations with more direct contact with older people tended to be the most frequently infected. Taken together, these findings suggested that strict infection control measures should be implemented, as well as education for the group of healthcare workers who have frequent contact with the elderly, because healthcare workers can initiate and spread the infections quickly among vulnerable groups.

The present study showed that the results of the SARS-CoV-2 IgG antibody test against anti-S antibodies from DENKA and Abbott generally matched and were almost in agreement with the RT-PCR-positive results, even approximately six months after natural infection. Meanwhile, the seropositivity of the anti-N antibody was relatively lower than that of the anti-S antibody. Recent literature suggests that IgG antibodies to the N protein decrease over time, while responses to the S protein are more stable over a longer period [28,29,30,31]. Besides, antibodies against the S protein is reportedly more specific than antibodies against the N protein due to lower cross-reactivity with other seasonal coronaviruses [32]. An impressive study investigated seropositivity patterns at different intervals (i.e., 2, 6, and 13 months) after an outbreak in the Lithuanian private sector in April 2020, when approximately one third of employees (94 out of 300) tested positive via RT-PCR [20]. This study showed that six months after the outbreak, 95.0% of 59 previously infected individuals had virus-specific anti-S antibodies, irrespective of the severity of infection, suggesting that specific antibodies persisted for longer than 6 months in the majority of cases, consistent with our results. 

In this present study, there were two individuals, a member of staff and a resident, who were RT-PCR negative but had positive anti-S and anti-N antibody titers. There is a possibility that this asymptomatic seropositive staff can be the source of infection, but other staff could be the source since 19 staff members declined to participate in this study, left the facility, or retired from it. Indeed, one extensive systematic review, including 34 studies in 2020, demonstrated a large unexplained false-negative of RT-PCR and suggested this was due to missed cases of asymptomatic infections (tau-squared = 1.39) [33,34]. Therefore, in addition to repeated RT-PCR testing, it is useful to conduct surveys with additional serological testing in cohorts of individuals to supplement RT-PCR results and capture asymptomatic cases missed by PCR, as in this study [34]. Those additional studies may give important information to clarify what the infection source was and what kind of infection controls should be taken to prevent future spread.

The findings in this report are subject to at least seven technical limitations. First, there is limited epidemiological information on the presumed cause of infection in the index case (i.e., the first resident who became ill on 10 November 2020) does not allow a detailed description of the transmission chains. Besides, there is incomplete epidemiologic information about potential visitors to associate with the case and limited information about interactions outside the nursing home that may have contributed to the initial virus entry. Altogether, this present study enrolled a small number of persons (i.e., 103) and was limited to describing their epidemiological characteristics and was unable to more objectively explore in depth the potential drivers associated with the exposure and transmission dynamics in this nursing home. Second, we were unable to perform viral genome analysis (e.g., whole genome sequencing), which, when combined with the lack of detailed epidemiologic information, makes it impossible to fully characterize the transmission patterns in this nursing home [35,36]. Three, it was not possible to collect epidemiological information on symptoms for each case, so the associations between RT-PCR testing results and symptomatic/asymptomatic symptoms could not be scrutinized. Importantly, previous research has suggested that infection during the pre-symptomatic period or from asymptomatic individuals may have been potential drivers in infection transmission within the facility, suggesting that they are likely to have contributed to transmission [37,38,39,40]. In particular, a study in a similar contextual setting to ours from Belgium suggests that approximately 14.0% and 50.0% of pre-vaccination seropositive staff and residents, respectively, did not report previous COVID-19 symptoms [41]. Fifth, the IgG antibody titer assay in this present study did not assess specific neutralizing antibody levels against SARS-CoV-2 owing to technical challenges, which may have underestimated exposure in the study population [42,43]. Future research will focus on more detailed quantifications of specific neutralizing antibody titer assays to explicitly conclude seroprevalence. Sixth, the present study only collected specific epidemiological data on the subjects (i.e., age, sex, and occupation for staff), which made it difficult to explicitly scrutinize the association between serum antibody titers and other crucial factors (e.g., comorbidities and anthropometric measurements). Though not available for this study, these factors could help disentangle the directionality of exact transmission and may be helpful for future studies. Finally, seroprevalence was estimated at a single time point approximately six months after natural infection, and no serology data were obtained soon after the outbreak, which limits generalizability. Ultimately, there remains room to examine the shift of antibody titers at multiple time points in the future.

Notwithstanding these limitations, the present study consistently demonstrated that the point pre-vaccination seroprevalence among the residents was higher compared to staff members in this outbreak in a nursing home in Niigata, Japan. Besides, the diagnostic performance in pre-vaccination residents and staff of a nursing home showed a relatively high match with RT-PCR results after approximately six months, partially highlighting that the anti-S IgG antibody tests may be useful as a diagnostic tool to scrutinize the possibility of COVID-19 infection. This present study highlights the value of serological analysis to understand the extent of SARS-CoV-2 circulation in this high-risk setting (e.g., long-term nursing homes), which also demonstrates the importance of repeatedly performing RT-PCR screening as an epidemic control measure for infectious diseases. Further studies are needed at the same institution to determine whether the post-vaccination anti-SARS-CoV-2 IgG antibodies, including neutralizing antibodies, are protective against the re-infection and, if so, the duration of protection among residents and staff.

## Figures and Tables

**Figure 1 viruses-14-02581-f001:**
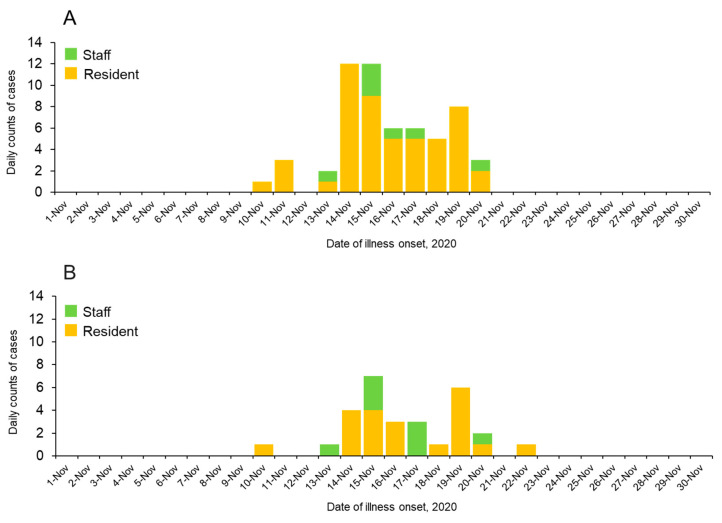
Epidemic histogram of SARS-CoV-2 outbreak in nursing care home in Niigata, Japan, November 2020. (**A**) Epidemic histogram of reverse transcription PCR (RT-PCR) positives at the time of the outbreak (*n* = 58). (**B**) Epidemic histogram of RT-PCR positives who participated in this study (*n* = 29). Daily counts of confirmed cases by RT-PCR tests are described as a function of the day of illness onset. Note that five persons whose date of illness onset by active epidemiological investigations was unknown were excluded in (**A**). Yellow and green bars correspond to staff and residents, respectively.

**Figure 2 viruses-14-02581-f002:**
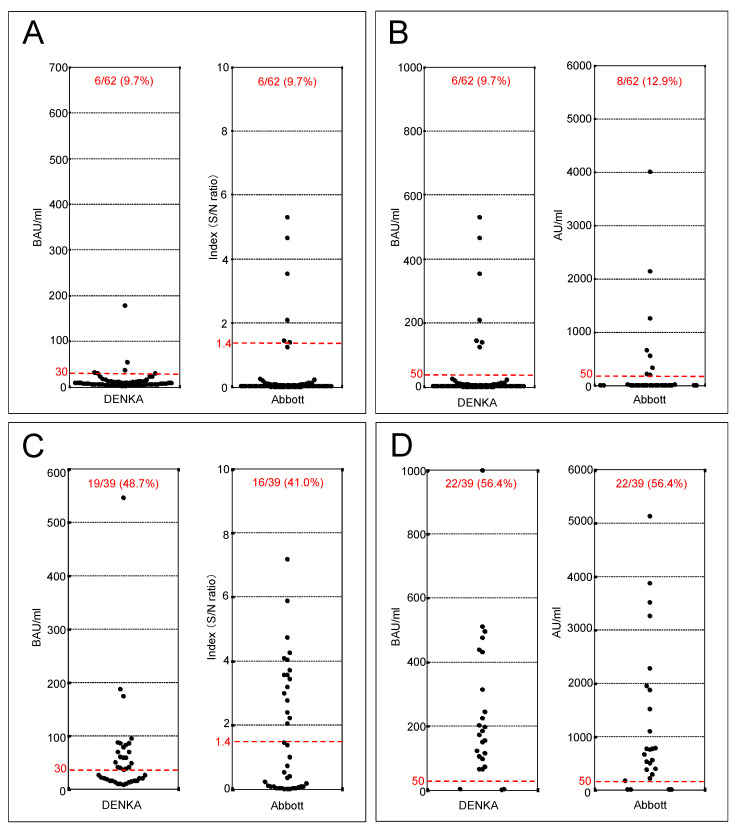
IgG antibody responses of SARS-CoV-2 in nursing care home in Niigata, Japan, April 2021 (*n* = 101). (**A**) Responses of severe acute respiratory syndrome 2 (SARS-CoV-2) in staff’s anti-nucleocapsid (N) IgG antibodies by DENKA (Tokyo, Japan) and Abbott (Chicago, IL, USA). (**B**) Responses of SARS-CoV-2 in staff’s anti-spike (S) IgG antibodies by DENKA and Abbott. (**C**) Responses of SARS-CoV-2 in resident’s anti-N IgG antibodies by DENKA and Abbott. (**D**) Responses of SARS-CoV-2 in resident’s anti-S IgG antibodies by DENKA and Abbott. The percentage of positive cases (%) is described and the red dotted line indicates the threshold for positivity (i.e., positive cutoff index).

**Table 1 viruses-14-02581-t001:** Epidemiological characteristics of the study population (*n* = 103).

Characteristic	Staff (*n* = 62, 60.1%)		Resident (*n* = 41, 39.9%)	
RT-PCR Test Result	Positive (*n* = 6, 9.7%)	Negative (*n* = 56, 90.3%)	Positive (*n* = 24, 58.5%)	Negative (*n* = 17, 41.5%)
Age (years), median (IQR)	34.0 (30.0–44.0)	50.0 (38.8–56.0)	90.0 (86.0–93.0)	93.0 (86.0–97.0)
Sex, *n* (%)				
Male	2 (33.3)	8 (14.3)	0 (0.0)	4 (23.5)
Female	4 (66.7)	48 (85.7)	24 (100.0)	13 (76.5)
Occupation, *n* (%)				
Doctor	0 (0.0)	4 (7.1)	NA	NA
Nurse	0 (0.0)	14 (25.0)	NA	NA
Caregiver	6 (100.0)	31 (55.4)	NA	NA
Clerk	0 (0.0)	7 (12.5)	NA	NA

Abbreviations: RT-PCR, reverse transcription PCR; SD, standard deviation; IQR, interquartile range; NA, not available. Notes: Data are displayed as median (interquartile range [IQR]) or *n* (%).

**Table 2 viruses-14-02581-t002:** Test seropositivity and seronegativity of the DENKA and Abbott methods for anti-N and anti-S IgG antibodies among patients with COVID-19 at the nursing home in Niigata, Japan in April 2021 (*n* = 101).

**DENKA (Tokyo, Japan)**	**Anti-N IgG Antibody**	**Anti-S IgG Antibody**
Seropositivity (%)	66.7 (20/30)	90.0 (27/30)
Seronegativity (%)	93.0 (66/71)	97.2 (69/71)
**Abbott (Chicago, IL, USA)**	**Anti-N IgG Antibody**	**Anti-S IgG Antibody**
Seropositivity (%)	66.7 (20/30)	90.0 (27/30)
Seronegativity (%)	97.2 (69/71)	95.8 (68/71)

Abbreviations: N, nucleocapsid; S, spike; IgG, immunoglobulin G. Notes: Seropositivity and seronegativity of the DENKA and Abbott methods were calculated using the results of the antibody titer as reference.

## Data Availability

Anonymized datasets generated during the study are available on request from the corresponding author.

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
