# Peer review of "Assessing the Pre-Vaccination Anti-SARS-CoV-2 IgG Seroprevalence among Residents and Staff in Nursing Home in Niigata, Japan, November 2020"

_viruses, 2022, doi:10.3390/v14112581_

Round 1
Reviewer 1 Report
I wished if you had presented the comorbidities and the anthropometrics of the participants and correlated your findings with them.

Reviewer 2 Report
Wagatsuma et al. measured SARS-CoV-2 antibodies in a nursing home six months after having an outbreak. They measure anti-spike and anti-nucleocapsid antibodies with two methodologies and found that half of the residents had antibodies against SARS-CoV-2.
The authors made a big effort to measure antibodies against SARS-CoV-2 with two methodologies to confirm their findings. I suggest the authors to confirm the concordance of their results with a Cohen's kappa. Nevertheless, the manuscript doesn’t provide anything new.
Reviewer 3 Report
Dear Authors,
The Manuscript entitled: “Assessing the Pre-vaccination Anti-SARS-CoV-2 IgG Seroprevalence among Residents and Staff in Nursing Home in Niigata, Japan, November 2020” has been reviewed.
The revised manuscript deserves attention since it highlights an important subject related to the dosage of the specific natural immune response, mainly anti-N and anti-S IgG, against SARS-CoV-2 infection in residents and staff of a Japanese Nursing Home located in Niigata. In fact, this dosage was done after six month of an COVID-19 outbreak in November 2020 in this Nursing Home. Results of this work are very important and showed that serological test, especially anti-S IgG tests, can be used for the detection of SARS-CoV-2 infection.
In general, the manuscript is well prepared, well written in English, and the discussion is rich in information related to the different results present in this study.
Please find below my remarks and comments concerning your manuscript:
(Minor Queries/Comments)
1- In the Abstract, line 22, authors are invited to explain when the 30 positive cases and 71 negative cases were tested by RT-PCR. It is clear in the manuscript that they were tested during the outbreak, but it is not very clear in the abstract.
2- In the Introduction section, line 39, authors are invited to replace “16 January 2020” by “January 16th, 2020”.
3- In the whole manuscript, especially regarding the Figure 1, authors are invited to replace the term “Curve” by the term “Histogram”.
4- In the Results section, line 169, authors are invited to replace “Yellow green and orange” and “Yellow and Green”.
5- In the Discussion section, line 264, authors are invited to put “et al.,” in italic.
Round 2
Reviewer 2 Report
The authors improved the manuscript adding Cohen's Kappa.
The manuscript improved overall. Nevertheless, I still think it should be published elsewhere.